# Cognitive Impairments, Neuroinflammation and Blood–Brain Barrier Permeability in Mice Exposed to Chronic Sleep Fragmentation during the Daylight Period

**DOI:** 10.3390/ijms24129880

**Published:** 2023-06-08

**Authors:** Clementine Puech, Mohammad Badran, Alexandra R. Runion, Max B. Barrow, Kylie Cataldo, David Gozal

**Affiliations:** 1Child Health Research Institute, Department of Child Health, School of Medicine, University of Missouri, 400 N Keene St., Suite 010, Columbia, MO 65201, USA; c.puech@health.missouri.edu (C.P.); mbadran@health.missouri.edu (M.B.); mbbhww@mail.missouri.edu (M.B.B.); kmhgg3@health.missouri.edu (K.C.); 2Undergraduate Student Research Program, University of Missouri, Columbia, MO 65201, USA; arrxh7@mail.missouri.edu; 3Department of Medical Pharmacology and Physiology, School of Medicine, University of Missouri, Columbia, MO 65201, USA

**Keywords:** sleep fragmentation, obstructive sleep apnea, explicit memory, inflammation, blood–brain barrier, cognition, microglia

## Abstract

Obstructive sleep apnea (OSA) is a chronic condition characterized by intermittent hypoxia (IH) and sleep fragmentation (SF). In murine models, chronic SF can impair endothelial function and induce cognitive declines. These deficits are likely mediated, at least in part, by alterations in Blood–brain barrier (BBB) integrity. Male C57Bl/6J mice were randomly assigned to SF or sleep control (SC) conditions for 4 or 9 weeks and in a subset 2 or 6 weeks of normal sleep recovery. The presence of inflammation and microglia activation were evaluated. Explicit memory function was assessed with the novel object recognition (NOR) test, while BBB permeability was determined by systemic dextran-4kDA-FITC injection and Claudin 5 expression. SF exposures resulted in decreased NOR performance and in increased inflammatory markers and microglial activation, as well as enhanced BBB permeability. Explicit memory and BBB permeability were significantly associated. BBB permeability remained elevated after 2 weeks of sleep recovery (*p* < 0.01) and returned to baseline values only after 6 weeks. Chronic SF exposures mimicking the fragmentation of sleep that characterizes patients with OSA elicits evidence of inflammation in brain regions and explicit memory impairments in mice. Similarly, SF is also associated with increased BBB permeability, the magnitude of which is closely associated with cognitive functional losses. Despite the normalization of sleep patterns, BBB functional recovery is a protracted process that merits further investigation.

## 1. Introduction

Obstructive sleep apnea (OSA) is characterized by an increased upper airway resistance and collapsibility during sleep which prompt the occurrence of obstructive events that result in intermittent hypoxia (IH) and episodic arousal events, i.e., sleep fragmentation (SF) [1,2]. OSA is associated with a markedly increased risk of a vast array of end-organ morbidities, including cardiovascular and metabolic diseases, as well as cognitive dysfunction [3,4,5,6,7,8,9,10,11,12,13]. It has become apparent that endothelium is one of the major end-organ targets of both IH and SF [14,15,16,17,18,19,20], and consequently OSA-related physiological perturbations can adversely affect the endothelial cell structure and function of the blood–brain barrier (BBB) [21,22,23,24,25,26,27,28,29].

The BBB is a multicellular, highly regulated barrier between the blood circulation and the central nervous system (CNS) [30,31] and plays a critical role in preserving brain homeostasis by regulating the exchange of substances between blood and the CNS. The BBB dynamic barrier is constitutively composed of an arrangement of capillary endothelial cells connected by tight junctions (TJs) and surrounded by the end feet of surrounding astrocytes and pericytes, which enable communication with neurons and microglia [31,32]. This uniquely arranged capillary network prevents the entrance of many substances, such as toxins and immune cells, from the blood while allowing and regulating the selective passage of essential nutrients and oxygen. Disruption of this regulated system results in disrupted regional or generalized brain homeostasis manifesting as activation and propagation of inflammatory cascades along with neurological deficits [33]. In the context of sleep, clear evidence has pointed to the emergence of increased permeability of the BBB in the context of acute sleep deprivation and restriction or chronic insufficient sleep, along with the increased activity of proinflammatory transcription factors such as NF-κB and increased levels of interleukins and TNF-α [34,35,36,37,38,39,40,41,42,43,44]. Thus, evidence of BBB disruption has been tied to an increased risk of neurodegenerative diseases and reduced cognitive abilities [45,46,47].

OSA has been closely linked to cognitive dysfunction, including impaired attention, memory, and executive functioning [4,10,11,12,48,49]. In this setting, murine experiments employing either IH or SF have clearly demonstrated that these hallmarks of OSA can impair both declarative memory (memory for facts and events) and procedural memory (memory for skills and habits) [50,51,52,53,54,55,56]. Thus, a potential mechanism underlying the relationship between SF and cognitive dysfunction may involve the disruption of the BBB. Accordingly, the objectives of the present study were to assess potential associations between BBB permeability, neuroinflammatory markers and cognitive function in mice chronically exposed to SF patterns. In addition, we also wished to examine whether such changes, if present, are reversible with cessation of SF exposures.

## 2. Results

As mentioned above, mice were exposed to both 4 weeks (SF4w) and 9 weeks (SF9w) of SF to evaluate whether there were any differences between mid-term and long-term SF effects on sleep, NOR or BBB permeability.

SF exposures for 4 weeks or 9 weeks increased the percentage of time spent in sleep during the dark period when compared to SC (SF4w: 40.3 ± 5% vs. 23.7 ± 4%, *p* < 0.0001 Figure 1A, SF9w: 39 ± 8% vs. 19.68 ± 4%, *p* < 0.0001, Figure 1B). No significant differences in sleep pattern were detected in SF9w and SF4w, and therefore results were pooled together (Figure 1C).

Two weeks after discontinuation of the SF-exposures in SF4w-exposed mice, the percentage time spent in sleep during the dark period remained significantly higher (30.1 ± 3%) in comparison to time in controls SC mice (23.2 ± 3%, *p* = 0.0011 Figure 1D). Sleep patterns returned to control levels in SF4w mice only after 6 weeks of sleep recovery (Figure 1E).

The preference scores in the NOR test were markedly decreased in mice after both 4 weeks (30.4 ± 15.5%) and 9 weeks of SF exposures (33 ± 32%) compared to their time-control counterparts (SC4w: 67.6 ± 32.92%, *p* = 0.0262 vs. SF4w; Figure 2A, and SC9w: 63.2 ± 15.4%, *p* = 0.0281 vs. SF9w; Figure 2B). Since no differences were apparent between SF4w and SF9w, results were pooled (Figure 2C). One mouse was excluded in the SC4w group due to becoming uncooperative during NOR. NOR preference scores were improved (41.2 ± 23%) vs. time controls (69.2 ± 18, *p* = 0.028- Figure 2D) but only normalized after 6 weeks of recovery (SF: 59.2 ± 12% vs. SC: 70.2 ± 16, *p* > 0.05 vs. time controls Figure 2E).

BBB permeability was markedly increased in SF4w-exposed mice (2054 ± 396 ng of dextran/g brain) when compared to SC4w mice (1168 ± 263 ng of dextran/g brain, *p* = 0.0004; Figure 3A). BBB permeability remained elevated after 9 weeks of SF (SF9w: 1573 ± 174 ng of dextran/g brain, vs. SC9w: 803.9 ± 89 ng of dextran/g brain, *p* < 0.0001; Figure 3B). No significant differences in BBB permeability were detected in SF9w and SF4w, which were pooled together (Figure 3C). Interestingly, no improvements in BBB permeability occurred after 2 weeks of sleep recovery in SF4w mice (1527 ± 183 ng of dextran/g brain vs. SC time control mice (808 ± 88 ng of dextran/g brain, *p* = 0.0262; Figure 3D). However, BBB permeability values returned to control levels in SF4w mice after 6 weeks of sleep recovery (Figure 3E).

To evaluate any potential association between BBB and cognitive functioning, BBB permeability values for each mouse were plotted against corresponding NOR performance and revealed a markedly robust inverse correlation (r Spearman = −0.8027; *n* = 29; Y = −1.825x + 7.219 *p* < 0.0001; Figure 3F).

Interleukin 1β levels were increased in SF4w mice (22.4 ± 6.7 pg/mL) when compared to SC4w mice (10.5 ± 3.2 pg/mL; *p* = 0.0189; Figure 4A). Similarly as previously shown [43], SF4w induced significant increases in brain TNF-α concentrations (2.37 ± 0.3 pg/µg brain vs. SC4w: 1.56 ± 0.2 pg/µg brain; *p* = 0.049; Figure 4B). In parallel, transcriptional activity of NF-κB was markedly enhanced in SF4w (0.86 ± 0.1 A.I.) when compared to SC4w controls (0.44 ± 0.03 AI, *p* = 0.0065; Figure 4C).

To further assess structural changes in BBB integrity, immunostaining of SF4w-exposed mouse brains revealed enhanced ionized calcium-binding adaptor molecule 1 (Iba-1), immunoreactivity indicative of activated microglia and primarily surrounding astrocytes and capillaries, as well as disrupted and reduced Claudin 5 expression patterns along endothelial cells within capillary structures further suggesting altered tight junctions (Figure 5).

## 3. Discussion

Sleep is a fundamental physiological function involved in the restoration of cellular and organ homeostasis and is essential for optimal daytime functioning [28,29]. Voluntary or imposed sleep deprivation or restriction has been extensively associated with neurobehavioral and cognitive deficits [22,50,57,58,59,60], and several studies have suggested that such sleep deficits lead to BBB dysfunction and activation of inflammatory processes within the CNS, as illustrated by increased levels of pro-inflammatory cytokines and microglial activation [40,43,61,62,63,64].

In this study, we showed that fragmented sleep induces inflammatory processes in the brain along with the structural and functional deterioration of BBB tight junctions, all of which are accompanied by a proportionate deficit in explicit memory function, as illustrated by the NOR performance. Furthermore, the recovery of BBB permeability by allowing SF-exposed mice to return to undisturbed sleep exhibits an asymmetric extinction trajectory requiring a more extended period than the original duration of SF.

Under normal circumstances, microglia exist in a quiescent state, with a highly ramified morphology and low levels of activation. However, in response to several stressors such as SF, microglia can become activated and undergo both morphological and functional changes [64,65]. Once microglia are activated, a cascade of inflammatory pathways is initiated, as illustrated by increased transcriptional activity of NF-κB and downstream secretion of cytokines such as IL-1β and TNF-α. These responses result in the disruption of BBB integrity with dysfunction of endothelial cells and structural perturbations in tight junctions and their constitutive elements, e.g., claudin 5, ultimately resulting in increased BBB permeability [40,62,63]. These processes can persist and even be amplified in a positive feedback loop, whereby microglia activation leading to increased inflammation and BBB disruption in turn leads to propagation and expansion of microglia activation [66,67]. Once the BBB is compromised, immune cells and potentially harmful substances such as toxins and pathogens can enter the brain, leading to incremental inflammation and brain cellular damage and apoptosis, a phenomenon that pertains to many neurodegenerative disorders, such as Alzheimer disease and Parkinson disease [21,24,45,68,69,70].

We have previously shown that chronic SF mimicking the sleep disruption that characterizes moderate to severe patients with OSA induces increased oxidative stress in the brain that culminates in cognitive and behavioral deficits in mice [43,50,55,71]. Furthermore, subsequent studies further indicated that SF exposures induced increases in TNF-α levels in CNS, which were critically involved in the increased sleep propensity that accompanies SF despite the absence of sleep deprivation in this experimental paradigm [43]. Indeed, administration of TNF-α neutralizing antibodies or transgenic inactivation of TNF receptors abrogated the sleep-promoting effects of SF [43]. Here, we provide further confirmatory evidence of such neuroinflammatory processes, in light of the increases in upstream transcriptional activation of NF-κB and increased brain levels of pro-inflammatory cytokines. Of note, these processes have also been described in the context of sleep deprivation in both humans and experimental animal models [62,72,73,74].

Sleep disturbances have been previously associated with increased BBB permeability and have been putatively ascribed to neuroinflammation [36,42,75]. Furthermore, our study, among others, has shown that the serum of patients suffering from OSA, possibly related to heightened levels of or pro-coagulant activity or alternatively that circulating plasma exosomes of children with OSA, increase the permeability and reduce the overall impedance in an in vitro model of BBB consisting of an endothelial cell monolayer [24,27,28,29]. Additional studies have also pointed to the critical role played by astrocytes in the changes in BBB barrier function, whereby increased expression of specific differentially expressed genes induced by sleep deprivation result in disruptions of BBB permeability [61,76]. In the current study, we found compelling evidence for in vivo increases in BBB permeability evoked by exposures to chronic SF, and further documented that the changes in BBB permeability are stable and do not progress with extended prolongation of the SF exposures. It will be of interest in future studies to evaluate the temporal characteristics (number of consecutive days) of BBB permeability changes in the context of SF, and also to determine the critical duration (hours/day) and cycling frequency (number of arousals per hour) that elicit BBB dysfunction.

The concurrent presence of cognitive deficits as illustrated by reduced explicit memory function in the NOR test and BBB endothelial dysfunction are likely correlated in the context of a disease such as OSA [14]. Indeed, earlier observations in children with OSA indicated a remarkable degree of overlap between the presence or absence of cognitive deficits and endothelial dysfunction [14]. Furthermore, Palomares and colleagues showed the presence of altered BBB function, as illustrated by water exchange characteristics across a proton gradient, in adult patients with OSA [77]. Our results indicate, for the first time, the uniquely robust association between the magnitude of the alterations in BBB permeability and the degree of cognitive dysfunction induced by chronic SF. Furthermore, the stability of the changes in BBB when SF was extended from a duration of 4 weeks to 9 weeks was accompanied by the stability of NOR performance rather than resulting in the continued progression and deterioration of NOR performance decrements. In the present study, we examined components of the neurovascular unit (NVU), i.e., the complex structure of brain microvascular endothelial cells, pericytes and astrocytes that results in the BBB under the leading assumption that perturbations in the NVU by SF would result in cognitive deficits. Indeed, the NVU is increasingly recognized as a major determinant of pathological CNS processes leading to cognitive and mood disorders (for a recent review, see: Ref. [78]). SF was associated with altered tight junction proteins, such as Claudin 5 (Figure 5). Previous studies have indicated that single nucleotide polymorphisms and differential methylation patterns of Claudin 5 underlie components of BBB integrity and function and are also associated with cognitive dysfunction in humans [79,80]. In mice, stress exposures will lead to the loss of Claudin 5, while transcriptional repression of Claudin 5 results in BBB hyperpermeability [81,82]. Similarly, we explored the expression changes in Iba1 with SF exposures. Iba1 is a cytoplasmic protein that is considered a pan-microglial marker, and its expression levels are tightly correlated with microglial activation and inflammation [83,84], while also playing a role in cognitive processes [85,86]. Our immunostaining findings showing an increased Iba1 expression are aligned with the expression of increased inflammatory markers and may further play a role in both BBB and cognitive detrimental effects induced by SF.

The delayed cognitive recovery and that of BBB integrity after cessation of SF4w exposures were somewhat anticipated. Indeed, previous work by Kopp and colleagues showed that mild short sleep restriction was accompanied by specific alterations in long-term potentiation and long-term depression as related to subunit composition of synaptically activated NMDA receptors and that sleep recovery reversed this process [87,88]. However, the recovery kinetics to long-term sleep restriction have not been extensively explored. In young people exposed to chronic sleep restriction, marked inhomogeneity responses became apparent in the recovery processes with some manifesting early recovery while others showing no evidence of recovery at all [89]. Furthermore, the relationships between OSA and cognitive dysfunction are complex, and although treatment of OSA with CPAP or other therapies has been shown to improve cognitive function [21,90,91,92,93,94,95], not all studies have corroborated such findings [96,97]. Furthermore, recovery is not systematically present and some patients may still have cognitive dysfunction after 6 months of CPAP treatment [4,49,98]. To this effect, in another murine model of OSA, we showed that the degree of reversibility of NOR performance was dependent of the antecedent duration of exposures suggesting that the longer the exposure the less likely full recovery would ensue [99]. Although in the current study we did not specifically explore the reversibility of explicit memory dysfunction induced by SF, the fact that even after 4 weeks of SF full recovery is possible is reassuring.

Studies in rodents on BBB permeability changes upon the termination of sleep restriction or deprivation have yielded quite conflicting results [35,36,40]. Importantly, it would seem that the recovery of the permeability of the BBB differs across different regions of the brain. For example, allowing for brief periods of sleep opportunity after 10 days of sleep deprivation was accompanied by a progressive recovery of the BBB except in regions such as the hippocampus and cerebellum [36]. It is likely that the differences between such findings and current study results can be explained by the different types of exposures. Indeed, animals were consistently exposed to sleep deprivation during a relatively short period of time (maximum 10 days), while chronic SF was implemented for 4 weeks in our experiments. Since the issue of BBB recovery is emerging as an important facet related to cognitive decline in the context of sleep perturbations, systematic exploration of the kinetics of such recovery and the underlying mechanisms, including the restoration of the NVU and the reversal of microglial activation, appears warranted [66].

Before we conclude, several limitations of the present study deserve to be mentioned. For obvious reasons related to the large number of mice that would be required, expanding our experiments to include more exposure durations and recovery times would have been impractical prior to completing current observations. In addition, we evaluated only male mice who were also relatively young when exposed to SF. Consequently, the effects of SF exposures in female, ageing, or very young mice remain unexplored. Finally, we did not specifically investigate a putative mechanism mediating the deleterious effects of SF on BBB and cognition. We exclusively explored the effects of SF exposure, and as such, assessment of the perturbations induced by other characteristics of OSA, such as intermittent hypoxia, episodic hypercapnia, and increased intrathoracic pressure swings and their combination, will have to await future studies.

In summary, chronic SF exposures mimicking the fragmentation of sleep that occurs among moderate to severely affected patients with OSA elicit cognitive impairments in mice. Similarly, SF is associated with increased BBB permeability along with the presence of inflammation in the brain, and the magnitude of BBB dysfunction is closely associated with the degree of deterioration in cognitive function. In addition, both BBB and cognitive dysfunction undergo complete recovery after cessation of SF exposures, but such processes appear to require extended periods of time, a finding that may be of translational clinical relevance in light of the markedly heterogenous cognitive improvement responses to therapy in OSA patients.

## 4. Materials and Methods

All experiments were approved by the Institutional Animal Care and Use Committee (IACUC) of the University of Missouri (IACUC 9720). All efforts were made to minimize animal suffering and to reduce the number of animals.

Seventy male C57BL/6 J mice (8-week-old) were purchased from Jackson Laboratory (Bar Harbor, ME, USA). Animals were housed in a controlled environment with 12 h light–dark cycles (6 a.m. to 6 p.m.) at constant temperature (24 ± 0.2 °C) with ad libitum access to food (normal chow) and water. All animals were allowed to recover and fully acclimate within the animal care facility for at least seven days after arrival. Animals were then randomly allocated into 2 different experimental groups, namely sleep fragmentation (SF) or sleep control (SC).

### 4.1. Sleep Fragmentation (SF)

The SF device used to induce sleep disruption has been extensively described in previous studies [43,50,52,55,58,100,101]. Briefly, mice were housed in custom-designed cages containing an automatic horizontal bar sweeping immediately above the cage floor from one side to the other side of the cage. The sweeper was operated by a nearly silent mechanically motorized system (Model 80391; Lafayette Instruments, Lafayette, IN, USA). To implement SF patterns that mimic OSA arousal patterns, 2 min intervals between each sweep (i.e., 30 events/h) were applied during the murine preferential rest period (i.e., daylight hours—6 a.m. to 6 p.m.) for a total period of 4 weeks or 9 weeks. To mimic resolution of OSA as achieved by optimal CPAP treatment, mice were exposed to SF for a total of 4 weeks and then the sweeper was discontinued as in SC conditions for a period of 2 or 6 weeks (Figure 6). Of note, initial experiments during the early validation of this experimental paradigm did not detect any evidence of systemic stress in the mice exposed. However, it should be pointed out the stress-related measures were not obtained in current experiments [56].

### 4.2. Sleep Recording

Sleep/wake activity was monitored using a validated, computerized piezoelectric system (PiezoSleep; Signal Solutions, Lexington, KY, USA). This noninvasive system, automatically scores sleep and waking states in mice (SleepStat; Signal Solutions, Lexington, KY, USA) [58,102,103,104,105,106]. Briefly, a piezoelectric film able to detect pressure variations is placed under the cage floor. The Piezoelectric system has been previously validated and exhibits ~90% accuracy compared to EEG/EMG-based sleep recordings and scoring systems [106,107].

### 4.3. Novel Object Recognition (NOR) Test

The novel object recognition (NOR) test was performed by three blind operators in a random order. All experiments were recorded from a vertical point of view with a video camera suspended above the experimental area. Between each trial, the experimental set up was cleaned with ethanol 70% to prevent odor cues.

The NOR test has been previously extensively described [50,58]. Briefly, this experimental paradigm is used to evaluate explicit memory based on the innate tendency for mice to explore novelty settings. For each trial, mice were placed in the center of a blue opaque open field plastic chamber. The NOR test is composed of 3 distinct phases. The habituation phase allows mice to explore the open field for 10 min. During the second phase of 5 min, two identical objects are placed in the arena. During the third and last phase, one of the objects is replaced by a new object. Different objects of different colors, shape and sizes were used as novel objects and placed randomly either on the left or on the right side of the arena. The mice were allowed to freely explore the objects for 5 min. Positive exploration by the mouse was defined as touching the object with the nose. The time spent exploring the objects was analyzed and quantified by the tracking software (Noldus Ethovision XT16 Software, Leesburg, VA, USA).

The total exploration time for both objects was recorded. Results were reported as preference score using the following formula [50,58,108,109]:$$Preference\;score=\frac{\mathrm{Time}\;\mathrm{spent}\;\mathrm{near}\;\mathrm{to}\;\mathrm{novel}\;\mathrm{object}}{\mathrm{Time}\;\mathrm{spent}\;\mathrm{near}\;\mathrm{to}\;\mathrm{all}\;\mathrm{objects}}\times 100$$

Mice who did not explore objects were removed from the experiment. Animals were considered to have an explicit preference for novelty if their preference score was >50%.

### 4.4. Inflammatory Markers

After the completion of SF or SC exposures, mice were euthanized by the introduction of 100% carbon dioxide into a euthanasia chamber. Then, brains were immediately dissected, flash frozen in liquid nitrogen and stored at −80 °C until assayed. The transcriptional activity of NF-κB (Active motif kit; Carlsbad, CA, USA) and TNF-α levels (Sinobiological Labs; Beijing, China) were evaluated in the frontotemporal cortical sections with commercially available kits according to manufacturer instructions. IL-1β expression levels were assayed with a commercial kit from Abcam (Cambridge, UK).

### 4.5. Blood Brain Permeability

Under deep isoflurane anesthesia, 3–5 kDa dextran–FITC (10 mg/mL- Sigma-Aldrich, St. Louis, MO, USA) suspended in PBS was injected intravenously in the mouse tail [110,111,112]. After 20 min, mice were euthanized with CO_2_, their blood was collected, and mice were then extensively perfused with PBS-heparin-20U to wash out any residual blood from the cerebral circulation. After perfusion, brains were removed, weighed and homogenized in tris-HCL (1 M, pH 7.5, 2:1 *v*/*w*) on ice. After centrifugation (10,000× *g* at 4 °C for 15 min), equal volumes of methanol were added to the supernatants, followed by another centrifugation (10,000× *g* at 4 °C for 15 min). Fluorescence in the supernatants (triplicate samples per mouse brain) was then detected at 528 nm with an excitation wavelength of 485 nm using a microplate reader [113]. Dextran concentration was calculated from a calibration curve, and tissue fluorescence values were normalized against fluorescence readouts from mouse brains without dextran.

### 4.6. Immunohistochemistry

Mice were euthanized and were transcardially perfused with PBS containing heparin-20U to wash out residual blood from cerebral circulation. Mice were further perfused with 4% PFA in PBS, and post-fixed in 4% PFA overnight at 4 °C. To prepare frozen sections, post-fixed brains were immersed for 2–3 days in 20% (*w*/*v*) sucrose in PBS for cryoprotection. Then, brains were sectioned in two and each portion was placed in a cryomold and embedded in OCT (Tissue-Tek optimal cutting Temperature, Thermo Fischer, Hanover Park, IL, USA) frozen in an isopentane dry ice bath. Brains were sectioned in 7 µm thickness sections on a Leica CM1850 cryostat, mounted in superfrost glass slides and frozen at −80 °C until assayed. Slides were dried at room temperature for 30 min, then placed on an epitope retrieval solution immersed in boiling water for 3 min and cooled for 5 min, and then subjected to immunostaining.

First, the non-specific binding sites were blocked for 2 days at 4 °C in a blocking solution constituted of 1% goat serum, 10% donkey serum and 1% BSA in PBS. Then, the first primary antibody (Table 1) was diluted in 1% BSA in PBS and incubated overnight at 4 °C in a humid atmosphere. The slides were then washed 3 times for 5 min in PBS at room temperature. The first secondary antibody (Table 1) was diluted in 1% BSA PBS and incubated at room temperature in humid atmosphere during 2 h and washed 3 times during 5 min in PBS. The same protocol was then repeated for the second and third couple of primary-secondary antibodies. The last wash was performed with water and the slides were then covered with mounting medium followed by the placement of a cover slide. Slides were then sealed with nail polish. Sections were then imaged using an epifluorescence microscope (Echo Revolve microscope; RVL-2-K2, Echo, San Diego, CA, USA).

### 4.7. Statistical Analysis

Statistical analysis was performed using Prism 9.2 for windows (GraphPad Software (Version 9), San Diego, CA, USA, www.graphpad.com (accessed on 1 April 2023)). Mann–Whitney tests were used to compare treatments in SF and SC groups. The data are expressed as mean ± SD. A two-tailed *p*-value < 0.05 was considered statistically significant.

## Figures and Tables

**Figure 1 ijms-24-09880-f001:**
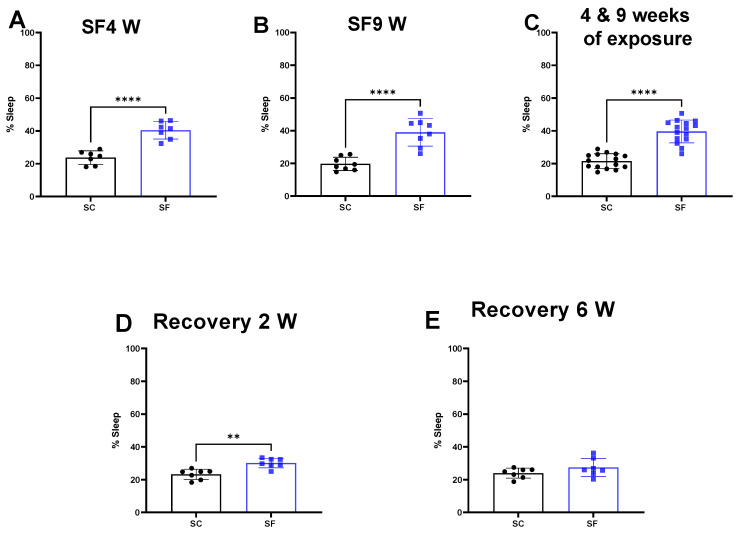
Sleep percentage after 4 weeks (panel (**A**)) or 9 weeks (panel (**B**)) or both together (panel (**C**)). After 4 weeks of SF, animals were placed in SC condition for 2 weeks (panel (**D**)) or 6 weeks (panel (**E**)). Data are shown as mean ± SEM, *n* = 7–8/for each group except for SF4w and SF9w. *n* = 14–15/group. ** *p* < 0.01, **** *p* < 0.0001.

**Figure 2 ijms-24-09880-f002:**
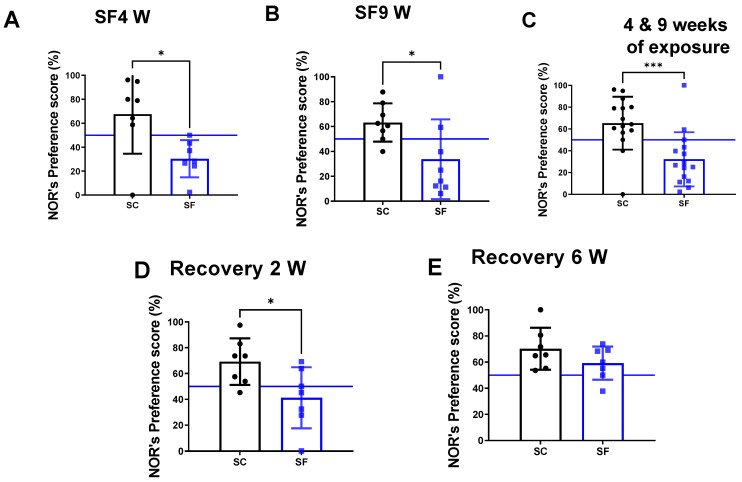
Explicit memory evaluated by NOR’s preference score test after 4 weeks (panel (**A**)) or 9 weeks (panel (**B**)) or both together (panel (**C**)). After 4 weeks of SF, animals were placed in SC condition for 2 weeks (panel (**D**)) or 6 weeks (panel (**E**)). Data are shown as mean ± SEM, *n* = 7–8/for each group except for SF4w and SF9w. *n* = 14–15/group. * *p* < 0.05, *** *p* < 0.001.

**Figure 3 ijms-24-09880-f003:**
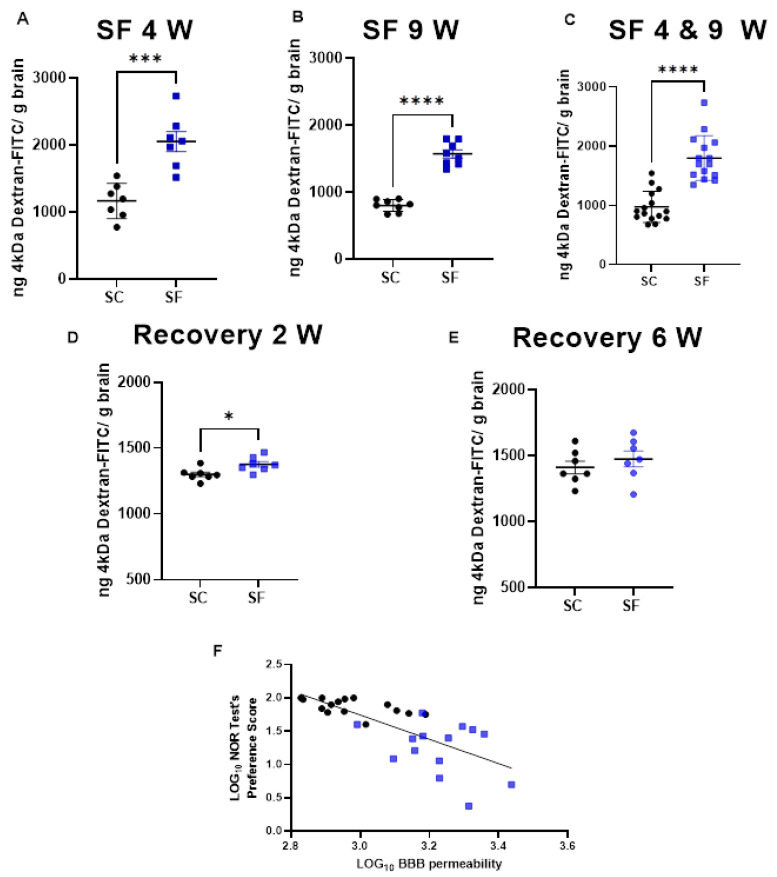
BBB permeability which was measured by systemic administration of 4 kDa-dextran-FITC is shown after 4 weeks (panel (**A**)) or 9 weeks (panel (**B**)) or both together (panel (**C**)). BBB permeability was also evaluated 2 weeks (panel (**D**)) and 6 weeks (panel (**E**)) after cessation of SF exposures in mice being subjected to SF for 4 weeks. Data are represented as mean ± SEM, *n* = 7/group. * *p* < 0.05, *** *p* < 0.001, **** *p* < 0.0001. Correlation between NOR test preference score and BBB permeability (panel (**F**)). *n* = 29, r Spearman = −0.8027, Y = −1.825x + 7.219, *p* < 0.0001.

**Figure 4 ijms-24-09880-f004:**
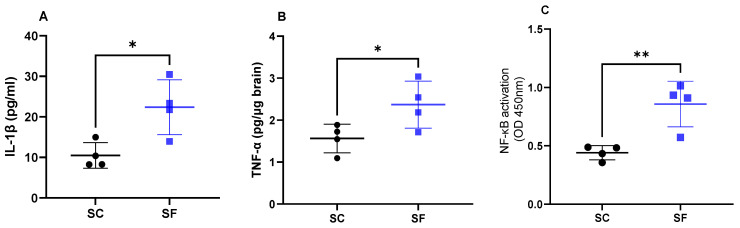
Inflammatory markers in the brain after 4 weeks of sleep fragmentation during daylight hours. IL-1 β (panel (**A**)), TNF-α (panel (**B**)) and NF-κB (panel (**C**)). Data are shown as mean of triplicates for each mouse ± SEM, *n* = 4. * *p* < 0.05, ** *p* < 0.01.

**Figure 5 ijms-24-09880-f005:**
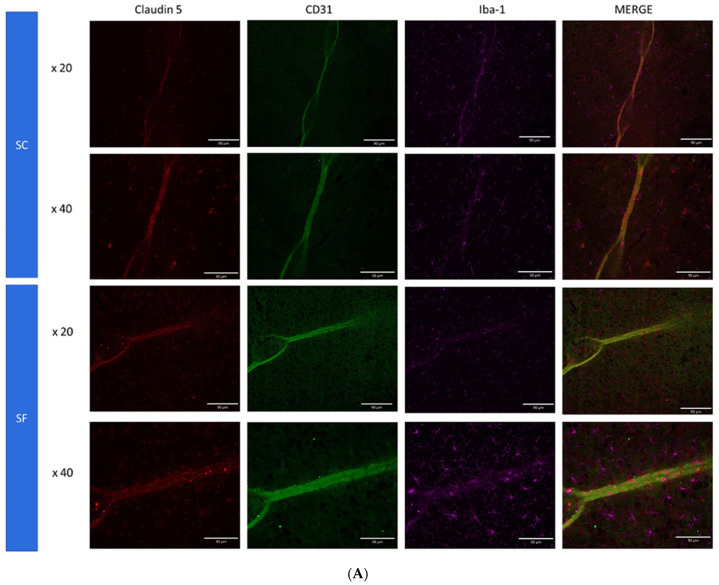
Triple immunostaining of cortical brain sections after 4 weeks of SF or SC exposures. Panel (**A**): Claudin 5 (red); Iba1 (purple) and CD31 (green); Panel (**B**): Claudin 5 (red); Iba1 (purple) and GFAP (green). The immunoreactivity of Claudin 5 is reduced and discontinuous while Iba1 expression is augmented along with increased astrocyte density in SF conditions.

**Figure 6 ijms-24-09880-f006:**
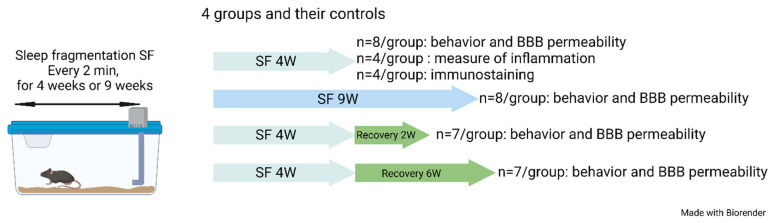
Schematic of the experimental design. Four groups of mice and their controls were used in this study.

**Table 1 ijms-24-09880-t001:** Primary and secondary antibodies used in immunostaining procedure and the sequence order of their application.

**Primary Antibodies**	**Dilution**	**Manufacturer**	**Order**
Rabbit monoclonal anti Claudin 5	1/1000	Abcam (Cambridge, UK) Ab131259	1
Rat monoclonal anti Iba1	1/500	Invitrogen (Waltham, MA, USA) MA5-38266	2
Goat Polyclonal anti-CD31/PECAM	10 μg/mL	Novus Biological (Englewood, CO, USA) AF3628	3 (panel 1)
Chicken polyclonal anti GFAP	1/5000	Invitrogen (Waltham, MA, USA) PA1-10004	3 (panel 2)
**Secondary Antibodies**	**Dilution**	**Manufacturer**	**Order**
Donkey anti-Rabbit Alexa fluor 555	1/1000	Thermofischer (Waltham, MA, USA) A32794	1
Donkey anti-Rat Alexa fluor 647	1/1000	Thermofischer (Waltham, MA, USA) A48272	2
Donkey anti-Goat Alexa fluor 488	1/1000	Thermofischer (Waltham, MA, USA) A32814	3 (panel 1)
Donkey anti Chicken Alexa fluor 488	1/5000	Jackson ImmunoResearch (West Grove, PA, USA) 703-545-155	3 (panel 2)

## Data Availability

Data pertaining to these studies are available from the corresponding author if and when such request is deemed appropriate and justified.

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
