# Peer review of "Cognitive Impairments, Neuroinflammation and Blood–Brain Barrier Permeability in Mice Exposed to Chronic Sleep Fragmentation during the Daylight Period"

_ijms, 2023, doi:10.3390/ijms24129880_

Round 1

Reviewer 1 Report

Obstructive sleep apnoea is an important medical problem. A better understanding of its mechanisms and clinically relevant consequences are important challenges.

Comments:

1: It is not clear from the current description of materials and methods how sleep quality and duration were monitored in mice. No EEG control was performed? Fragmented sleep was simulated during daytime. What was the duration of sleep during the day in both groups?

2.      It is not clear why the response of the mice to stress was not evaluated, nor were adrenal hormone levels determined in the groups. Physiological indicators were not assessed in both groups, including cardiovascular system. It is not clear how the experiment affected the behavior of the mice, nutrition, and other important parameters that might have influenced the results of the study. How did these physiological parameters change over the course of the study?

3. it is recommended to make the course of the experiment more detailed.

4. the current experiment does not fully simulate obstructive sleep apnoea, because not only intermittent sleep but also intermittent hypoxia, which plays an important role in endothelial dysfunction and disruption of the blood-brain barrier, is important for the disease. In addition, the mechanisms of awakening in obstructive sleep apnoea and in the current experiment are different.

Reviewer 2 Report

Problems in editing: results are before methods.

Different fonts.

Reviewer 3 Report

The study of Clementine Puech and co-authors focused on BBB permeability, neuroinflammatory markers after so-called sleep fragmentation model in young adult mice. I have some serious concerns about experimental design and methodology.

I am confused by the model of sleep fragmentation.
Page 9: "we showed that fragmented sleep in the absence of curtailment of sleep duration ..... ". Parameters of sleep were not measured here, the authors referred to the literature, which describes this technique (i.e., doi: 10.1186/1742-2094-9-91). In my opinion, the authors should prove that sleep was indeed fragmented, and sleep duration was not reduced.

Why "simulating the recurrent arousals that characterize obstructive sleep apnea"? I am not convinced that "2-min intervals between each sweep (i.e., 30 events/h)" would "implement SF patterns" that "mimic OSA arousal patterns".

Are there any proofs that "horizontal bar sweeping immediately above the cage" floor caused obstructive sleep apnea?

What brain structures were investigated? Figure 4 demonstrates "cortical brain sections". This is an important issue, because BBB permeability and neuroinflammatory markers might differ in different brain areas.

It is well known that "permeability at the BBB is dynamically controlled by circadian rhythms and sleep" [doi: 10.1016/j.tins.2019.05.001 and many other reports]. The BBB permeability is a dynamic process, and it is higher during deep sleep [i.e., doi: 10.1016/j.tins.2019.05.001]. Have the factors influencing BBB permeability been taken into account? What was behavioral state of mice when they were taken for dextran injections and perfusion?

Why was the NOR test used to examine memory? This test is known to characterize the differences in the exploration time of novel and familiar objects. The authors cited the brilliant review 10.1007/s10339-011-0430-z , but did not explain their motivation for using NOR test instead of other tests, such as spatial learning in a Morris water maze, instrumental or associative learning. The short- and long-term memory tests seem to better characterize cognitive functions (the aim of this study).

In this study, the NOR test was repeated at leat three times. When was the first NOR test performed? Are there any corrections for repeated measures?

Page 12: "Mice who did not explore objects were removed from the experiment." How many animals were excluded? Control or experimental groups?

Perhaps the schema of experiments showing the number of mice would help to understand the study design. How many mice were used for each test? It follows that only 7-8 subjects per group were tested for NOR.

Minor.

Figure 4. The plots are too dark

Round 2

Reviewer 1 Report

The authors added more information and made some comments. However, some questions remain:

1. The authors showed that previous studies excluded the possibility of stress development in mice and changes in their physiological and biochemical markers. On the other hand, the duration of the current exposure to sleep fragmentation differs significantly from the examples given. It must be justified that the current duration of experimental exposure (4-9 weeks) in mice also does not affect the development of stress.

2.      Why were the 2-minute intervals chosen for sleep fragmentation? Did the mice manage to fall asleep during these intervals? Or was it daytime sleep deprivation? What is the rationale for using 2-minute wake intervals to simulate fragmented sleep in OSA?

Author Response

Open Review

Quality of English Language

( ) I am not qualified to assess the quality of English in this paper
( ) English very difficult to understand/incomprehensible
( ) Extensive editing of English language required
( ) Moderate editing of English language
( ) Minor editing of English language required
(x) English language fine. No issues detected

Yes

Can be improved

Must be improved

Not applicable

Does the introduction provide sufficient background and include all relevant references?

( )

(x)

( )

( )

Are all the cited references relevant to the research?

( )

(x)

( )

( )

Is the research design appropriate?

( )

( )

(x)

( )

Are the methods adequately described?

( )

( )

(x)

( )

Are the results clearly presented?

( )

( )

(x)

( )

Are the conclusions supported by the results?

( )

( )

(x)

( )

Comments and Suggestions for Authors

The authors added more information and made some comments. However, some questions remain:

  1. The authors showed that previous studies excluded the possibility of stress development in mice and changes in their physiological and biochemical markers. On the other hand, the duration of the current exposure to sleep fragmentation differs significantly from the examples given. It must be justified that the current duration of experimental exposure (4-9 weeks) in mice also does not affect the development of stress.

  - The reviewer is correct that we did not specifically obtain measures of systemic stress in mice exposed to our sleep fragmentation paradigm and rather relied on our previous data that was assessed during shorter exposures. We have now made a comment to this effect in the paper.  

  1. Why were the 2-minute intervals chosen for sleep fragmentation? Did the mice manage to fall asleep during these intervals? Or was it daytime sleep deprivation? What is the rationale for using 2-minute wake intervals to simulate fragmented sleep in OSA?

 - The rationale for 2 min intervals during daylight hours corresponding to the usual sleep period of mice was selected to mimic a moderate to severe sleep apnea condition in which the obstructive events lead to about 30 arousals per hour of sleep. During chronic sleep fragmentation, there is no sleep restriction or deprivation such that mice actually the same amounts as controls over the 24-h period, i.e., similar to what happens in patients with sleep apnea.

Submission Date

14 April 2023

Date of this review

19 May 2023 07:45:12

Open Review

Quality of English Language

(x) I am not qualified to assess the quality of English in this paper
( ) English very difficult to understand/incomprehensible
( ) Extensive editing of English language required
( ) Moderate editing of English language
( ) Minor editing of English language required
( ) English language fine. No issues detected

Yes

Can be improved

Must be improved

Not applicable

Does the introduction provide sufficient background and include all relevant references?

( )

( )

(x)

( )

Are all the cited references relevant to the research?

( )

( )

(x)

( )

Is the research design appropriate?

( )

( )

(x)

( )

Are the methods adequately described?

( )

(x)

( )

( )

Are the results clearly presented?

( )

(x)

( )

( )

Are the conclusions supported by the results?

( )

( )

(x)

( )

Comments and Suggestions for Authors

The authors answered some of my questions and added some information in their manuscript. 

  1. Introduction is short and lacks focus. I highly recommend that to clearly identify the aims of the study and appropriately formulate the hypothesis to be tested.

 - with all due respect, the Introduction is succinct as all introductions for manuscript should while containing the important and salient issues leading to the driving hypothesis and aims of the study.

  1. It appears that they used PiezoSleep Mouse Behavioral Tracking System https://www.sigsoln.com/portfolio-item/piezosleep-mouse-behavioral-tracking-system/. This approach must be described in more details in methods. This fantastic devise provides huge raw data. Please explain what data were of your interest?

 - The description of the device and methodology is comprehensive and addresses all of the requested information by the reviewer. For more details regarding validation and development of the machine-learning algorithms related to this approach we strongly recommended going to the source papers.

  1. The major problem is the absence of a solid hypothesis. Introduction  obstructive sleep apnea and "physiological perturbations" caused by OSD, but they did not use an appropriate mice model. The experimental group was housed "in custom-designed cages containing an automatic horizontal bar sweeping immediately above the cage floor from one side to the other side of the cage."  Do there exist any concrete evidence that this housing in these extreme conditions causes sleep fragmentation? This has to be explained in the manuscript.

    4. The device is encased in a normal mouse cage identical to every other mouse cage in any vivarium. SF device is not meant to induce obstructive sleep apnea; it is mimicking only the sleep fragmentation that occurs in patients with sleep apnea. The validity and relevance to sleep apnea of the sleep fragmentation paradigm as developed by our laboratory more than 15 years ago has been extensively demonstrated – please see the references cited in the paper. Furthermore, this approach is not extensively used by many other investigators around the world.

I am not convinced that the results can be attributed to the obstructive sleep apnea. "The horizontal bar sweeping immediately above the cage" floor should affect behavior, and sleep fragmentation is just a part of it. 

We never contended that sleep apnea was being imposed on the mice; the paradigm elicits sleep fragmentation.

What cortical areas were taken for the analysis of neuroinflammatory markers? This is an important issue, because neuroinflammatory markers might differ in different brain areas.

 - this is clearly indicated in the methods

  1. Why was the NOR test used to examine memory? 

 - this test examines declarative memory and is less susceptible to repetition bias.

  1. In this study, the NOR test was repeated at least three times. When was the first NOR test performed? Are there any corrections for repeated measures?

 - These tests were performed at the end of SF exposure, 2 weeks and 6 weeks later in the animals delineated in the methods.

  1. One mouse was excluded in control group after 4 weeks of exposure.

Now  mentioned in the Results.

Reviewer 3 Report

The authors answered some of my questions and added some information in their manuscript. 

1. Introduction is short and lacks focus. I highly recommend that to clearly identify the aims of the study and appropriately formulate the hypothesis to be tested.

2. It appears that they used PiezoSleep Mouse Behavioral Tracking System https://www.sigsoln.com/portfolio-item/piezosleep-mouse-behavioral-tracking-system/. This approach must be described in more details in methods. This fantastic devise provides huge raw data. Please explain what data were of your interest?

3. The major problem is the absence of a solid hypothesis. Introduction  obstructive sleep apnea and "physiological perturbations" caused by OSD, but they did not use an appropriate mice model. The experimental group was housed "in custom-designed cages containing an automatic horizontal bar sweeping immediately above the cage floor from one side to the other side of the cage."  Do there exist any concrete evidence that this housing in these extreme conditions causes sleep fragmentation? This has to be explained in the manuscript.

4. SF device is not meant to induce obstructive sleep apnea; it is mimicking only the sleep fragmentation that occurs in patients with sleep apnea

I am not convinced that the results can be attributed to the obstructive sleep apnea. "The horizontal bar sweeping immediately above the cage" floor should affect behavior, and sleep fragmentation is just a part of it. 

5. Inflammation-related studies were performed in cortical sections.

What cortical areas were taken for the analysis of neuroinflammatory markers? This is an important issue, because neuroinflammatory markers might differ in different brain areas.

6. Why was the NOR test used to examine memory? 

7.  In this study, the NOR test was repeated at least three times. When was the first NOR test performed? Are there any corrections for repeated measures?

8.  One mouse was excluded in control group after 4 weeks of exposure.

This should be mentioned in the Results.

9. Figure 6. The plots are too dark

Author Response

(The authors gave the same response as above.)

Round 3

Reviewer 1 Report

This is an interesting and potentially useful study. However, there are serious questions about the study design. Sleep quality was not adequately controlled. It is not known whether it was fragmented sleep or still sleep deprivation, as the 2-minute intervals were not experimentally validated. How the quality of sleep changed over the course of the experiment over the 9 weeks? There are no controls for physiological and biochemical markers of stress that may have affected the results of the study. In this regard, it is not clear what the induced BBB permeability abnormalities are related to. It is not clear how the levels of inflammatory markers in the brain and blood correlate.

Reviewer 3 Report

The answers I've received to my questions have been unclear and did not coincide with what I was asking for.
I do not see critical changes in the original text.

1. Introduction is short and lacks focus. I highly recommend that to clearly identify the aims of the study and appropriately formulate the hypothesis to be tested.

Response - with all due respect, the Introduction is succinct as all introductions for manuscript should while containing the important and salient issues leading to the driving hypothesis and aims of the study. - ???

2. It appears that they used PiezoSleep Mouse Behavioral Tracking System https://www.sigsoln.com/portfolio-item/piezosleep-mouse-behavioral-tracking-system/. This approach must be described in more details in methods. This fantastic devise provides huge raw data. Please explain what data were of your interest.

Response - The description of the device and methodology is comprehensive and addresses all of the requested information by the reviewer. For more details regarding validation and development of the machine-learning algorithms related to this approach we strongly recommended going to the source papers. -- ??? Please explain in detail what data were of your specific interest.

3. The major problem is the absence of a solid hypothesis. Introduction  obstructive sleep apnea and "physiological perturbations" caused by OSD, but they did not use an appropriate mice model.

???

4. The experimental group was housed "in custom-designed cages containing an automatic horizontal bar sweeping immediately above the cage floor from one side to the other side of the cage."  Do there exist any concrete evidence that this housing in these extreme conditions causes sleep fragmentation? This has to be explained in the manuscript.

Response -The device is encased in a normal mouse cage identical to every other mouse cage in any vivarium. SF device is not meant to induce obstructive sleep apnea; it is mimicking only the sleep fragmentation that occurs in patients with sleep apnea. The validity and relevance to sleep apnea of the sleep fragmentation paradigm as developed by our laboratory more than 15 years ago has been extensively demonstrated – please see the references cited in the paper. Furthermore, this approach is not extensively used by many other investigators around the world.

??? This has to be explained in more details in the manuscript.

What cortical areas were taken for the analysis of neuroinflammatory markers? This is an important issue, because neuroinflammatory markers might differ in different brain areas.

Response - this is clearly indicated in the methods

??? I cannot find this.
